# Sustained Surface ICAM-1 Expression and Transient PDGF-B Production by Phorbol Myristate Acetate-Activated THP-1 Cells Harboring Blau Syndrome-Associated *NOD2* Mutations

**DOI:** 10.3390/children8050335

**Published:** 2021-04-25

**Authors:** Mizuho Nishiyama, Hong-jin Li, Ikuo Okafuji, Akihiko Fujisawa, Mizue Ehara, Naotomo Kambe, Fukumi Furukawa, Nobuo Kanazawa

**Affiliations:** 1Department of Dermatology, School of Medicine, Wakayama Medical University, Wakayama 641-0012, Japan; mizmiz254-med@yahoo.co.jp (M.N.); lhj215@163.com (H.-j.L.); dajs@wakayama-med.ac.jp (F.F.); 2Department of Pediatrics, Kobe City Medical Center General Hospital, Kobe 650-0047, Japan; ikuoka@gmail.com; 3Department of Dermatology, Graduate School of Medicine, Kyoto University, Kyoto 606-8507, Japan; akihiko@kuhp.kyoto-u.ac.jp (A.F.); nkambe@kuhp.kyoto-u.ac.jp (N.K.); 4Department of Dermatology, Graduate School of Medicine, Chiba University, Chiba 260-8670, Japan; ehara@faculty.chiba-u.jp; 5Department of Dermatology, School of Medicine, Hyogo College of Medicine, Nishinomiya 663-8501, Japan

**Keywords:** Blau syndrome, *NOD2* mutation, ICAM-1, PDGF-B

## Abstract

Objectives: Blau syndrome is a distinct class of autoinflammatory syndrome presenting with early-onset systemic granulomatosis. Blau syndrome-causing *NOD2* mutations located in the central nucleotide-oligomerization domain induce ligand-independent basal NF-κB activation in an in vitro reporter assay. However, the precise role of this signaling on granuloma formation has not yet been clarified. Methods: Blau syndrome-causing *NOD2* mutations were introduced into human monocytic THP-1 cells, and their morphological and molecular changes from parental cells were analyzed. Identified molecules with altered expression were examined in the patient’s lesional skin by immunostaining. Results: Although the production of proinflammatory cytokines was not altered without stimulation, mutant *NOD2*-expressing THP-1 cells attached persistently to the culture plate after stimulation with phorbol myristate acetate. Sustained surface ICAM-1 expression was observed in association with this phenomenon, but neither persistent ICAM-1 mRNA expression nor impaired ADAM17 mRNA expression was revealed. However, the transient induction of PDGF-B mRNA expression was specifically observed in stimulated THP-1 derivatives. In the granulomatous skin lesion of a Blau syndrome patient, ICAM-1 and PDGF-B were positively immunostained in *NOD2*-expressing giant cells. Conclusions: Sustained surface ICAM-1 expression and transient PDGF-B production by newly differentiating macrophages harboring mutant *NOD2* might play a role in granuloma formation in Blau syndrome.

## 1. Introduction

Autoinflammatory diseases constitute a group of genetic disorders whose main clinical features are recurrent episodes of inflammatory lesions that can affect the skin, joints, bones, eyes, gastrointestinal tract and nervous system, in association with signs of systemic inflammation [1,2]. Blau syndrome is a distinct class of autoinflammatory syndrome showing early-onset systemic granulomatosis [3,4,5]. In comparison to sarcoidosis, which is a multi-organ granulomatous disease with unknown etiology, Blau syndrome is histologically undistinguishable but clinically distinguished by the triad of skin rash, uveitis and arthritis without apparent lung involvement.

Blau syndrome has been shown to be caused by heterozygous missense mutations in the *NOD2/CARD15/NLRC2* gene [3,6]. NOD2 is a member of the NOD-like receptor (NLR) family of molecules and is composed of two amino-terminal caspase recruitment domains (CARDs), a centrally located nucleotide-binding oligomerization domain (NOD) and carboxy-terminal leucine-rich repeats (LRR). This molecule is expressed intracellularly in antigen-presenting cells (APC) and recognizes muramyl dipeptide (MDP), the minimum common component of bacterial cell wall peptidoglycan, to form a complex with CARD-containing serine/threonine kinase, RICK, and to induce immune responses through nuclear factor (NF)-κB activation [4,7]. NF-κB is one of the most important transcription factors inducing the expression of various cytokines, growth factors and cell adhesion molecules, and plays critical roles in developmental/survival and inflammatory processes [8]. Thus, NOD2 functions as an intracellular sensor for bacterial invasion. While mutations in the LRR impairing MDP-dependent NF-κB activation are reportedly associated with Crohn’s disease (CD), Blau syndrome-associated *NOD2* variants are localized in the NOD and show increased MDP-independent basal NF-κB activation, which is measured by an in vitro reporter assay using HEK293 cells [3,9,10,11]. By analysis of nine types of *NOD2* mutations collected from 20 Japanese Blau syndrome patients, the basal NF-κB activity level, which was presented as a ratio of the NF-κB activity in the absence of MDP to that in the presence of MDP, was shown to range from 0.3 for the E383G mutation to 0.9 for the N670K mutation, compared to 0.05 for the wild-type [12]. Although a clear correlation was reportedly absent between clinical severity and the basal NF-κB activation level of the corresponding *NOD2* mutation, a tendency was observed when the subjects were limited to ocular complications in cases with frequent R334W and R334Q mutations [12]. Furthermore, the therapeutic effect for Blau syndrome of thalidomide, a potent immunomodulatory drug suppressing NF-κB activation, may also suggest the role of NF-κB in the pathogenesis of this disease [13].

Granulomatous reaction is a distinct pathological pattern of chronic inflammation forming granuloma, and is classified into several types: foreign body, suppurative, tuberculoid, palisaded, interstitial and sarcoidal [14]. Sarcoidal granuloma is considered to be caused by a super-delayed hypersensitivity reaction directed at any unknown antigen. The release of inflammatory mediators, especially interferon (IFN)-γ, from activated T helper (Th) 1 cells is considered to be indispensable for the activation and accumulation of macrophages and subsequent granuloma formation [15]. In the case of pulmonary sarcoidosis, underlying latent infection with some distinct microorganism, such as *Propionibacterium acnes* and *Mycobacterium species*, has been shown [16]. In addition, some foreign substance such as silica can be found in the case of scar sarcoidosis [17]. These facts suggest the role of infection and/or innate immunity in the development of sarcoidal granuloma.

As suggested by the results of in vitro experimental findings, constitutive NF-κB activation in NOD2-expressing APC is considered responsible for granuloma formation in Blau syndrome. However, the precise role of such NF-κB activation and subsequent changes are still undefined. Therefore, to better understand these issues, *NOD2* mutations caused by Blau syndrome were introduced into human monocytic THP-1 cells to generate another in vitro model resembling monocytes. Any alterations from parental cells were identified by morphological and molecular analyses. Furthermore, the expression of identified intercellular cell adhesion molecule (ICAM)-1 with a sustained surface expression and platelet-derived growth factor (PDGF)-B with a transiently-induced production by phorbol myristate acetate (PMA)-treated THP-1 derivatives was analyzed in the lesional skin of a Blau syndrome patient by immunostaining.

## 2. Materials and Methods

### 2.1. Cell Culture

THP-1 cells were originally purchased from ATCC (American Type Culture Collection, Manassas, VA, USA) and maintained in RPMI1640 supplemented with 10% fetal bovine serum as previously described [18]. Transfected THP-1 derivatives were maintained in the presence of 500 µg/mL G418 (Invitrogen, Carlbad, CA, USA) to avoid the excessive proliferation of wild-type cells.

### 2.2. Reagents and Antibodies

PMA was purchased from Sigma-Aldrich (St. Louis, MO, USA). Human recombinant interleukin (IL)-4 and macrophage-colony stimulation factor (M-CSF) were purchased from Peprotech (Rocky Hill, NJ, USA). Mouse anti-FLAG M2 (Sigma-Aldrich), ICAM-1 (Immunotech), tumor necrosis factor (TNF)α (Santa Cruz Biotechnology, Inc. Dallas, TX, USA), NOD2 and interferon (IFN)γ (eBioscience) monoclonal antibodies (mAbs) were purchased. Rabbit anti-cadherin-11 (Zymed), NOD2 (Sigma-Aldrich) and PDGF-B (Novus Biologicals) polyclonal Abs were also purchased.

### 2.3. Transfection

The complementary DNAs (cDNAs) for wild-type (WT), R334W and N670K mutant *NOD2* were digested from the corresponding plasmids in the p3xFLAG-CMV vector, which were previously used for the NF-κB reporter assay, and recloned into the pIRES2-EGFP vector (Clontech) [3]. Each of these plasmids and the mock vector were linearized with AflII (New England Biolabs, Ipswich, MA, USA) and introduced into THP-1 cells using Amaxa Nucleofactor (Lonza Cologne GmbH, Cologne, Germany) according to the manufacturer’s recommended protocol. The transfected cells were selected in the presence of 750 µg/mL of G418.

### 2.4. RT-PCR Analysis

Total mRNA was extracted from THP-1 derivatives using Sepasol (Nakarai Tesque, Kyoto, Japan), and 1 µg of the mRNA was applied for cDNA synthesis using the Superscript first-strand synthesis kit including reverse transcriptase II and poly-T primer (Invitrogen). PCR was performed using Ex-Taq (Takara Bio Inc., Otsu, Japan) and ABI2720 thermal cycler (Thermo Fisher Scientific, Waltham, MA, USA) under the following conditions: 95 °C for 5 min followed by 35 cycles of 94 °C for 30 s, 57 °C for 30 s and 72 °C for 1 min, and finally 72 °C for 10 min. Specific primer pairs are listed in Table 1. 

### 2.5. Flow Cytometry

THP-1 derivatives were stained with primary antibody or the corresponding isotype control and the subsequent appropriate secondary antibodies, and were provided for analysis with FACSCaliber and BD CellQuest^TM^ Pro software (BD Biosciences, San Jose, CA, USA).

### 2.6. Immunohistochemistry

10-µm-thick sections of formaldehyde-fixed paraffin-embedded skin biopsy specimens, which were obtained from a Blau syndrome patient with a R334W mutation, were deparaffinized and subjected to staining with mouse monoclonal anti-ICAM-1, TNFα, IFNγ or rabbit polyclonal anti-PDGF-B Abs or the control mouse IgG1 or rabbit immunoglobulin. Positive staining was visualized using the VECTASTAIN Elite ABC Kit (Vector Laboratories, Inc., Burlingame, CA, USA) and DAB substrate kit (Dako Denmark A/S, Glostrup, Denmark) according to the manufacturer’s protocol. The sections were counterstained with hematoxylin.

### 2.7. Double Immunofluorescence Staining

The Blau syndrome patient’s skin specimen was applied for double immunofluorescence staining with a set of rabbit polyclonal anti-NOD2 Abs and mouse monoclonal anti-ICAM-1 Abs or a set of mouse monoclonal anti-NOD2 Abs and rabbit polyclonal anti-PDGF-B Abs at 1:50 dilution. The specimen was then stained with Alexa Fluor 488-conjugated anti-rabbit IgG, Alexa Fluor 555-conjugated anti-mouse IgG and Hoechst 33342. Sections were mounted with Vectashield (Vector Laboratories, Inc.) and observed with confocal microscopy (LSM780, Carl Zeiss Microscopy, Jena, Germany).

## 3. Results

### 3.1. No Altered mRNA Expression of Proinflammatory Cytokines in Mutant NOD2-Expressing THP-1 Derivatives

THP-1 cells were transfected by electroporation with either mock, FLAG-tagged WT, R334W or N670K mutant *NOD2* cloned in the pIRES2-EGFP vector. Among colonies developing in the presence of G418 at the manufacturer’s recommended concentration (750 µg/mL), those expressing EGFP and intracellular FLAG were selected by flow cytometry and were subsequently analyzed with RT-PCR for *NOD2* mRNA expression. Although *NOD2* mRNA expression was only faintly observed in mock-transfected THP-1 cells, significant expression was observed in WT or mutant *NOD2*-transfected THP-1 derivatives. For each derivative, one representative colony showing an almost equal expression of *NOD2* mRNA was selected and used for further analyses (Figure 1). The predominant expression of the mutant *NOD2* in the corresponding colony was confirmed by direct sequencing of the RT-PCR product (data not shown). Without stimulation, the expression of mRNA for proinflammatory cytokines such as TNFα and IL-8 was similar for WT and mutant *NOD2*-expressing THP-1 derivatives (Figure 1).

### 3.2. Long-Term Attachment of Mutant NOD2-Expressing THP-1 Derivatives after PMA Stimulation

Following PMA stimulation, THP-1 cells are known to differentiate along the monocytic lineage and to acquire characteristics of mature macrophages, including a loss of proliferation and an increased HLA-DR expression [19,20]. A large difference was morphologically observed between WT and mutant *NOD2*-expressing THP-1 derivatives after PMA stimulation. As shown in Figure 2, all THP-1 derivatives were attached to the culture plate and spread pseudopods 24 h after PMA addition. However, on day 3, mock or WT *NOD2*-expressing THP-1 derivatives floated into the medium again and proliferated, whereas mutant *NOD2*-expressing THP-1 derivatives remained attached to the plate and spread more and longer pseudopods. Such characteristic features were still apparent on day 7, especially in the case of N670K mutant *NOD2*-expressing THP-1 derivatives, as made clear after washing (bottom column of Figure 2). In contrast, R334W mutant *NOD2*-expressing THP-1 derivatives floated into the medium again and proliferated on day 7.

### 3.3. Sustained Surface Expression of ICAM-1 on Mutant NOD2-Expressing THP-1 Derivatives

To explore the mechanism underlying the long-term attachment of mutant *NOD2*-expressing THP-1 derivatives after PMA stimulation, the surface expression levels of various adhesion molecules were analyzed by flow cytometry. As shown in Figure 3, the surface expression of ICAM-1 was initially upregulated in all THP-1 derivatives on day 2 and then decreased to almost the basal level in mock or WT *NOD2*-expressing THP-1 derivatives on day 6. However, the expression remained higher for mutant *NOD2*-expressing THP-1 derivatives at this time point, even when considering the significant fluorescence alteration of the isotype control.

### 3.4. No Remarkable Alteration of ICAM-1 or ADAM-17 mRNA Expression Underlies Sustained Surface Expression of ICAM-1

By RT-PCR analysis, ICAM-1 mRNA expression in all THP-1 derivatives was initially upregulated on day 2 and similarly decreased on day 6 after PMA addition, suggesting that the sustained surface expression of ICAM-1 was not due to prolonged mRNA expression (Figure 4a, left column). We then analyzed the mRNA expression of ADAM-17, which reportedly mediates the cleavage of the ectodomain of ICAM-1 [21]. However, ADAM17 mRNA was similarly expressed in all unstimulated THP-1 derivatives, and no further increase was observed in any of the derivatives after PMA addition, suggesting that a defective ICAM-1 cleavage by ADAM-17 mRNA induction was not involved in sustained surface ICAM-1 expression (Figure 4a, middle column).

### 3.5. Transient PDGF-B mRNA Expression in PMA-Stimulated Mutant NOD2-Expressing THP-1 Derivatives

To explore the mRNA specifically induced in PMA-stimulated *NOD2*-expressing THP-1 cells, RT-PCR was performed for various cytokines and growth factors. Among them, mRNA expression of PDGF-B, which was undetectable in all unstimulated THP-1 derivatives, was strongly induced in mutant *NOD2*-expressing THP-1 derivatives, especially in the case of the N670K mutant, but only weakly induced in mock or WT *NOD2*-expressing cells on day 2 after PMA addition (Figure 4b, left column). PDGF-B is an essential growth factor involved in wound healing and might play a pivotal role in granuloma formation through activation of the surrounding fibroblasts. Notably, the induced PDGF-B mRNA expression was not sustained and decreased to the basal level in all THP-1 derivatives on day 6 after PMA addition (Figure 4b, left column). In contrast, IL-8 mRNA expression was induced to a similar extent on day 2 and was sustained on day 6 after PMA addition in all THP-1 derivatives, as shown in the right column of Figure 4b.

### 3.6. ICAM-1 and PDGF-B Protein Expression in NOD2-Expressing Giant Cells in the Lesional Skin of a Blau Syndrome Patient

To explore the site of ICAM-1 and PDGF-B expression and its relationship with NOD2 expression in Blau syndrome lesions, ICAM-1 and PDGF-B protein expression in the lesional skin specimen of a Blau syndrome patient harboring the *NOD2* R334W mutation was examined immunohistochemically and by double-immunofluorescence staining with NOD2. By immunohistochemistry, ICAM-1 expression was observed linearly just beneath the surface of multinucleated giant cells (MGCs), while PDGF-B expression was observed diffusely within (cytoplamic or perinuclear) granuloma-forming epithelioid cells and MGCs, as shown in Figure 5a. In contrast, no expression of TNFα or IFNγ was detected in the lesional skin (Figure 5a). By double-immunofluorescence staining, ICAM-1 and PDGF-B were weakly stained in MGCs and were both well co-localized with NOD2, which was clearly stained in the perinuclear region of MGCs (Figure 5b,c).

## 4. Discussion

Blau syndrome is a rare but genetically and histologically distinct disease, characterized by a gain-of-function *NOD2* mutation and noncaseating epithelioid cell granuloma. Since monocytic cells with a *NOD2* mutation are considered to play an essential role in granuloma formation in Blau syndrome, THP-1 cells of a human monocytic lineage were selected for a cellular model to analyze the pathogenesis of this disease. Therefore, it was unexpected that THP-1 cells expressing R334W or N670K-mutant *NOD2* showed no upregulated production of TNFα or IL-8 (Figure 1). However, these results are consistent with a previous report showing normal or less responsiveness of Blau syndrome patients’ peripheral blood mononuclear cells to various PAMP molecules [22]. Subsequently, we tried PMA, a strong NF-κB activator, to activate THP-1 cells. PMA is also known to induce the differentiation of THP-1 cells into macrophage-like cells [19,20]. Indeed, all THP-1 derivatives changed their shape in a similar way immediately after PMA addition and differentiated into macrophage-like attaching cells. However, very surprisingly, mutant *NOD2*-expressing THP-1 derivatives further extended their pseudopods and remained attached to the plate at subsequent observations, whereas control THP-1 cells began to float in the medium again and started to proliferate (Figure 2). These results suggest that the activating stimulus created by PMA is transient and induces regulatory pathways in the later stage. Since the surface expression of ICAM-1 on THP-1 derivatives seems to be correlated with their differentiation stage into macrophage-like cells (as shown in Figure 3), the downregulation of the surface ICAM-1 expression might be associated with the PMA-induced regulatory pathway. Further, this PMA-induced regulatory pathway might be abrogated by the Blau syndrome-causing *NOD2* mutation. Although the alteration of mRNA expression due to NF-κB activation was expected, neither the mRNA of ICAM-1 itself or of its shedding enzyme ADAM-17 was significantly changed, as shown in Figure 4a. Notably, the putative correlation of the surface ICAM-1 expression and its mRNA expression remains to be elucidated more quantitatively. As the level of soluble ICAM-1 in bronchoalveolar lavage is reportedly correlated with the severity of sarcoidosis, ICAM-1 expression in monocytic cells seems to have some association with granuloma formation [23].

Another interesting point is the induction of PDGF-B, but not IL-8, mRNA in mutant NOD2-expressing THP-1 cells after PMA activation (Figure 4b). Although an exhaustive study would be required to clarify the specificity of PDGF-B induction, the specific expression of PDGF-B was also observed immunohistochemically in the lesional skin of a Blau syndrome patient, as shown in Figure 5a. Since NF-κB is reportedly capable of inducing both PDGF-B and IL-8 mRNA expressions, some regulatory mechanism or other specific transcription factor might be involved [24,25]. PDGF-B is a strong activator of fibroblasts and plays an important role in tissue repair. Interestingly, PDGF-B expression was detected as early as day 1 and persisted for at least 14 days in the cutaneous tuberculin reaction as an in vivo human model of T cell-mediated delayed hypersensitivity [26]. As is the case with sarcoidosis, which can finally cause lung fibrosis, overexpression of PDGF-B in mouse lungs reportedly induced a complex phenotype that encompassed both features of emphysema and fibrosis [27]. Although the role of PDGF-B in granuloma formation is unclear, it might work on preparing the fibrous network surrounding granulomas.

To generate MGCs, various PAMP molecules or uric acid crystals were added to the PMA-stimulated THP-1 derivatives. However, no fusion of the attached cells was observed (data not shown). Although it was reported that exposure of Blau syndrome patients’ peripheral blood monocytes to IL-4 plus M-CSF caused their morphologic change into fibroblastic/dendritic cells with Langhans-type MGCs, neither fibroblastic/dendritic change nor MGC formation was induced in our THP-1 variants by this method [13].

Diffuse ICAM-1 expression along the surface of MGC and PDGF-B expression within granuloma-forming epithelioid cells and MGC formation in a Blau syndrome patient’s skin specimen seem compatible with their predicted functions (Figure 5a). In contrast, no expression of TNFα or IFNγ in the lesional skin was unexpected. Although the expression of both TNFα and IFNγ in the lesional skin of a Blau syndrome patient was previously reported, the genetic background of the patient was not clarified [28]. This suggests a distinct mechanism for granuloma formation in Blau syndrome, possibly independent of T cells. To examine this possibility, the mutant *NOD2*-expressing THP-1 variants were injected subcutaneously into athymic (nu/nu) mice with or without PMA. However, they never formed epithelioid cell granulomas with MGCs in vivo (data not shown). The co-localization of ICAM-1 and PDGF-B with NOD2 seems consistent with our experimental results (Figure 5b,c). However, it remains to be elucidated whether the expressed *NOD2* mRNA in MGCs is mutated because the patient is heterozygous to the mutation.

Collectively, these observations suggest that sustained surface ICAM-1 expression on and transient PDGF-B production in newly differentiating macrophages that harbor a mutant *NOD2* and respond to some stimuli might play a role in granuloma formation in Blau syndrome patients.

## Figures and Tables

**Figure 1 children-08-00335-f001:**
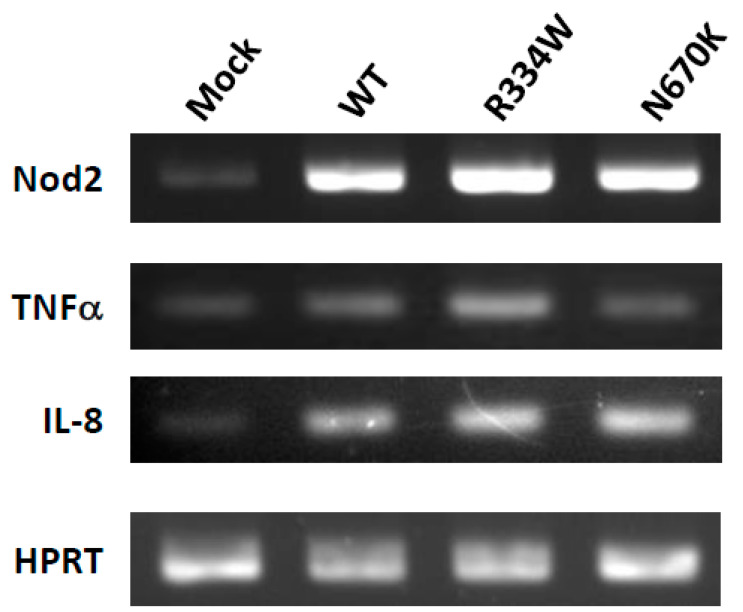
No altered mRNA expression of proinflammatory cytokines in mutant *NOD2*-expressing THP-1 derivatives. THP-1 cells were transfected with mock, WT, R334W or N670K mutant *NOD2* cDNA-containing letroviral vector and were selected for study in the presence of G418. From the surviving colonies, THP-1 derivatives with similar expression levels of *NOD2* mRNA revealed by RT-PCR were selected for further analyses (top row). Expression of TNFα or IL-8 mRNA in these derivatives was then analyzed by RT-PCR without stimulation. Hypoxanthine phosphoribosyltransferase (HPRT) was analyzed as a control for ubiquitous expression (bottom row).

**Figure 2 children-08-00335-f002:**
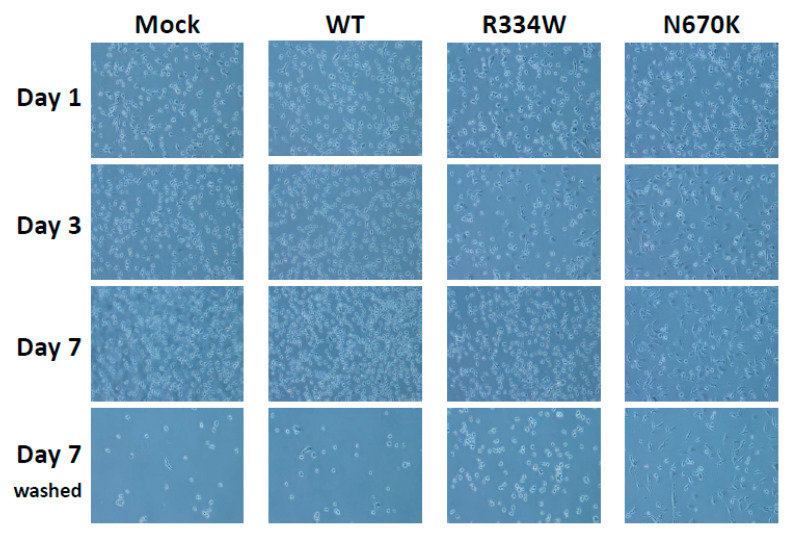
Long-term attachment of mutant *NOD2*-expressing THP-1 derivatives after PMA stimulation. THP-1 derivatives were stimulated with 10 mM of PMA, and photographs were taken under bright field lightning at the indicated time points. On day 7 after PMA addition, photographs were again taken after washing the floating cells (bottom row).

**Figure 3 children-08-00335-f003:**
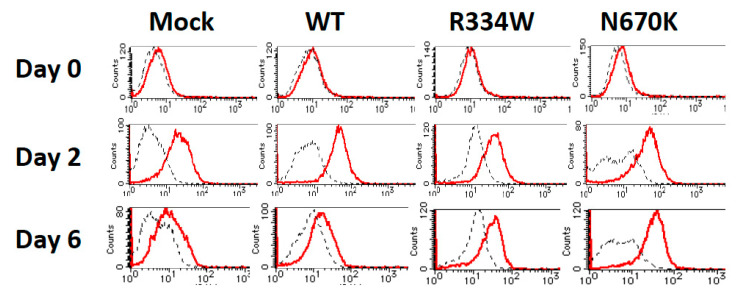
Sustained surface expression of ICAM-1 on mutant *NOD2*-expressing THP-1 derivatives. The surface expression of ICAM-1 on THP-1 derivatives was analyzed with flow cytometry at the indicated time points after PMA addition. The isotype control is shown by a dotted line.

**Figure 4 children-08-00335-f004:**
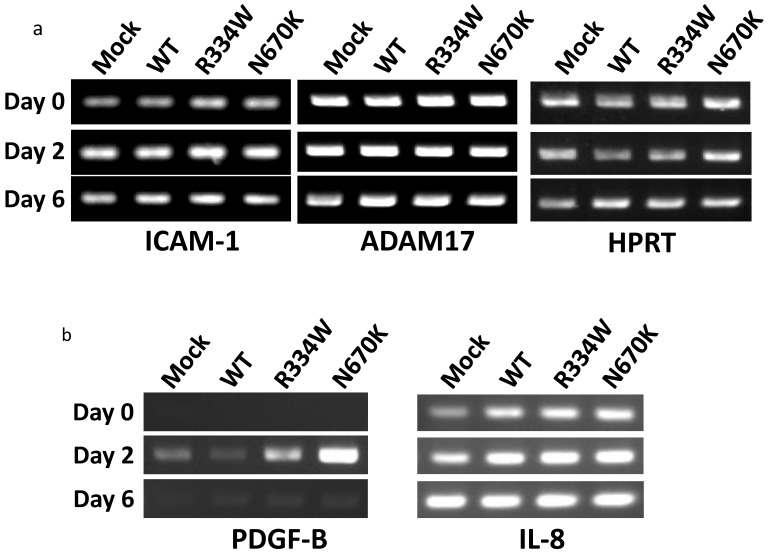
No altered ICAM-1 or ADAM-17 mRNA expression but a transient PDGF-B mRNA expression in PMA-stimulated mutant *NOD2*-expressing THP-1 derivatives. The expression of (**a**) ICAM-1 or ADAM-17 mRNA and (**b**) PDGF-B or IL-8 mRNA on THP-1 derivatives was analyzed by RT-PCR using the cells shown in Figure 3 and was compared between WT and mutant NOD2-expressing derivatives at each time point. No remarkable difference was observed in the expression of ICAM-1, ADAM17 or IL-8.

**Figure 5 children-08-00335-f005:**
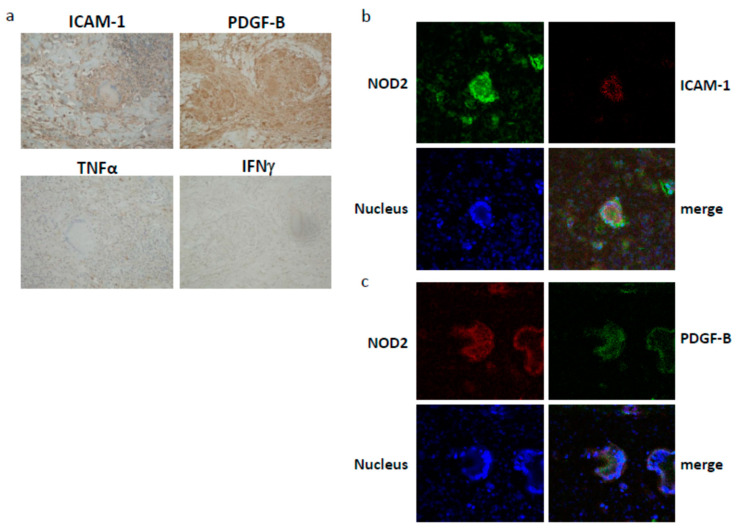
(**a**) Expression of ICAM-1, PDGF-B, TNFα or IFNγ in the lesional skin of a Blau syndrome patient with a *NOD2* R334W mutation was analyzed by immunohistochemistry. (Original magnification: ×100). (**b**,**c**) ICAM-1 and PDGF-B protein expression in *NOD2*-expressing giant cells in the lesional skin of a Blau syndrome patient. Expression of *NOD2* (green) with ICAM-1 (red) or of *NOD2* (red) with PDGF-B (green) was analyzed by double immunofluorescence and observed by confocal microscopy. Nuclei were stained by Hoechst 33342 and shown in blue. (Original magnification: ×200).

**Table 1 children-08-00335-t001:** Primer pairs used for RT-PCR analyses.

	Forward	Reverse
**NOD2**	AGACTCAGCTTCCCAAGGTCTG	AGAACACGTAGCAGCACATGCC
**IL-8**	AAGGAATAGCATCAATAGTGAGTTTG	GGACACAAGCTTAAACCCAGA
**ICAM-1**	CCTTCCTCACCGTGTACTGG	AGCGTAGGGTAAGGTTCTTGC
**ADAM17**	CCTTTCTGCGAGAGGGAAC	CACCTTGCAGGAGTTGTCAG
**PDGF-B**	CCTTTGATGATCTCCAACGC	GATCTTTCTCACCTGGACAG
**HPRT**	AATTATGGACAGGACTGAACGTC	CGTGGGGTCCTTTTCACCAGCAAG

## Data Availability

The data presented in this study are available on request from the corresponding author. The data are not publicly available due to patient’s privacy.

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
