# Peer review of "Sustained Surface ICAM-1 Expression and Transient PDGF-B Production by Phorbol Myristate Acetate-Activated THP-1 Cells Harboring Blau Syndrome-Associated NOD2 Mutations"

_children, 2021, doi:10.3390/children8050335_

Round 1
Reviewer 1 Report
The authors have performed a laboratory study looking ICAM-1 and PDGF-B production in Blau-type monocytic cells. This is a well performed study, with well described aims, methodology and hence the authors have made appropriate conclusions. The authors have elucidated a potential mechanism for the formation of granulomas in BS.
Author Response
Dear Reviewer 1,
Thank you very much for your positive comments.
Sincerely,
Nobuo Kanazawa
Reviewer 2 Report
General Comments: The investigators present preliminary results regarding the effects of Blau syndrome associated (BS) NOD2 mutations in transfected derivatives of human monocytic THP-1 cells. The study introduces a potential new model for BS signaling leading granuloma formation. Morphological and molecular changes were assessed. Interesting and potentially significant changes in attachment were described after stimulation with PMA. Although this phenomenon appeared to be associated with sustained expression of intercellular cell adhesion molecule, ICAM-1 in mutant-expressing THP-1 derivatives, neutralization with mAB to ICAM-1 did not affect attachment. The study includes examination of mRNA expression for other adhesion molecules, cytokines, and other biologically relevant activators to probe the signaling process. Critical supporting data such as cytokine production that would strengthen the results are mentioned but not shown. The findings and tentative conclusions are discussed.
Specific comments and questions:
- The model system requires further information on assay validation.Have the in vitro culture conditions for assays using THP-1 derivatives been optimized for cell density, passage time for NOD-2 activation, length of exposure to specific activators, and timing of different responses? Data showing dose response of derived THP-1 cells to a standard agonist such as MDP as the positive control compared to PMA would be helpful.
- Can the effects of PMA on cell differentiation be separated from activation? Are these events detectable and different in WT and mutant NOD2 cells?
- The data in Figure 1 show mRNA expression of WT and mutant NOD2 that is described as showing effective transfection. Cytokine mRNA and ELISA detection are described as “similar” in response to defined activators, but these critical data are not shown. Were the transfected THP-1 cells or the control THP-1 cells capable of being activated? Data should be included.
- Morphologic differences shown in Figure 2 are somewhat convincing. Additional data showing biological activity would be helpful and important for their hypothesis. The authors mention that PMA was much more effective at inducing inflammatory cytokines in THP-1 derivatives after 24 hours of stimulation than various PAMP molecules. This data including unstimulated controls should be shown.
- The investigators report that altered attachment to the tissue culture plate in mutant NOD2-expressing THP-1 after PMA activation showed some association with increased expression of mRNA for platelet-derived growth factor (PDGF) as well as (ICAM)-1 expression. Was this associated with cell activation?
- Additional proof reading and attention to text are needed. The Abstract sentence “Sustained surface ICAM-1 expression was observed in conjunction with this phenomenon, but sustained ICAM-1 expression or impaired ADAM17 mRNA expression was not observed” is self-contradictory. The intended ICAM should be supplied. The intended word “subjects” is given as “objects” in line 61, referring to ocular cases; also “stained” in line 125 for “stained”
Author Response
Dear Reviewer 2,
Thank you very much for your constructive comments.
I have revised the manuscript according to your comments.
Point-by-point answers to your comments are shown below.
- The model system requires further information on assay validation. Have the in vitro culture conditions for assays using THP-1 derivatives been optimized for cell density, passage time for NOD-2 activation, length of exposure to specific activators, and timing of different responses? Data showing dose response of derived THP-1 cells to a standard agonist such as MDP as the positive control compared to PMA would be helpful.
Answer: Culture of THP-1 cells and PMA treatment were performed according to Ref18. However, treatment of the cells with MDP and other PAMP molecules might not be fully optimized. To avoid misleading, parts of the experiments using PAMP molecules have been deleted.
- Can the effects of PMA on cell differentiation be separated from activation? Are these events detectable and different in WT and mutant NOD2 cells?
Answer: As the Reviewer pointed out, effects of PMA on cell differentiation and activation might be undistinguishable. The change of surface ICAM-1 expression and induction of PDGF-B mRNA expression might reflect cell activation.
- The data in Figure 1 show mRNA expression of WT and mutant NOD2 that is described as showing effective transfection. Cytokine mRNA and ELISA detection are described as “similar” in response to defined activators, but these critical data are not shown. Were the transfected THP-1 cells or the control THP-1 cells capable of being activated? Data should be included.
Answer: As previously described, parts of the experiments using PAMP molecules have been deleted to avoid misleading. The change of surface ICAM-1 expression and induction of PDGF-B mRNA expression might reflect cell activation.
- Morphologic differences shown in Figure 2 are somewhat convincing. Additional data showing biological activity would be helpful and important for their hypothesis. The authors mention that PMA was much more effective at inducing inflammatory cytokines in THP-1 derivatives after 24 hours of stimulation than various PAMP molecules. This data including unstimulated controls should be shown.
Answer: As previously described, parts of the experiments using PAMP molecules have been deleted to avoid misleading.
- The investigators report that altered attachment to the tissue culture plate in mutant NOD2-expressing THP-1 after PMA activation showed some association with increased expression of mRNA for platelet-derived growth factor (PDGF) as well as (ICAM)-1 expression. Was this associated with cell activation?
Answer: As the Reviewer pointed out, the induction of PDGF-B mRNA expression might reflect cell activation.
- Additional proof reading and attention to text are needed. The Abstract sentence “Sustained surface ICAM-1 expression was observed in conjunction with this phenomenon, but sustained ICAM-1 expression or impaired ADAM17 mRNA expression was not observed” is self-contradictory. The intended ICAM should be supplied. The intended word “subjects” is given as “objects” in line 61, referring to ocular cases; also “stained” in line 125 for “stained”
Answer: Thank you for the corrections. I have revised the sentences according to your comments.
I believe the manuscript has sufficiently been revised and is suitable for publication in Children.
Sincerely,
Nobuo Kanazawa
Round 2
Reviewer 2 Report
Overall comments: The revised paper is intended to introduce a new model based on introduction of BS-causing NOD2 mutations into THP-1 cells. The investigators present initial results from experiments using one colony from two THP-1 derivatives and wild type (WT) THP-1 cells and include related studies on ICAM-1 and PDGF-B protein expression in the lesional skin specimen of a BS patient harboring the NOD2 R334W mutation. In response to reviewer’s critique, the authors have modified the text. In particular they have removed some, but not all summarized data presented as “data not shown” which detracts from the presentation of their potentially novel studies. Other issues remain as mentioned below.
Specific issues:
- The Results describe that THP-1 NOD2-expressing derivatives differ from wildtype (WT) in response to PMA, specifically they state that the derivatives remained attached to the plate longer, with more and longer pseudo- pods, and were resistant to proliferation at Day 7, while WT cells were not attached and proliferated on Day 7. Data shown in Fig 2 show some possible differences in attachment between WT and mutant NOD2 THP-1 derivatives, but no data about proliferation. The authors should provide some data on proliferation resistance or modify the statement to exclude proliferation resistance.
- Data in Figure 3 are described as showing that surface expression of ICAM-1 was increased after PMA stimulation in all THP-1 derivatives on day 2 and then was nearly reduced to basal level in mock or WT NOD2-expressing THP-1 cells but not in mutant NOD2 THP-1 on day 6. Figure3 shows unevenly matched “y” axis scales in the panels and rows. Also, the marked fluorescent shift in mutant NOD2-expressing derivatives’ isotype controls on Day 6 is concerning and would limit the differences. The bottom row using overlay does not take into account the strong isotype shift especially for R334W.
- The authors mention in Results that they analyzed expression of other surface markers and also did experiments using neutralizing antibodies and provide an assessment of these as a summary of new results, but they do not show data for either. The data must be shown or could be made available as supplementary material but otherwise should not be included.
- The legend for Figure 4 should briefly describe the method and delineate the figure. The authors need to consider their flow data in Figure 3, address the isotype shift, and also explain how surface expression of ICAM-1 could be discordant with mRNA expression.
- Abstract: This sentence should be revised to make sense- “Sustained surface ICAM-1 expression was observed in conjunction with this phenomenon, but neither of sustained ICAM-1 mRNA expression or impaired ADAM17 mRNA expression was not observed.”
Author Response
Dear Reviewer 2,
Thank you very much again for your further constructive comments.
I have re-revised the manuscript according to your comments.
Point-by-point answers to your comments are shown below.
- The Results describe that THP-1 NOD2-expressing derivatives differ from wildtype (WT) in response to PMA, specifically they state that the derivatives remained attached to the plate longer, with more and longer pseudo- pods, and were resistant to proliferation at Day 7, while WT cells were not attached and proliferated on Day 7. Data shown in Fig 2 show some possible differences in attachment between WT and mutant NOD2 THP-1 derivatives, but no data about proliferation. The authors should provide some data on proliferation resistance or modify the statement to exclude proliferation resistance.
Answer: Unfortunately, our data in Fig 2 lacked quantitative analyses. To avoid misleading, statement on proliferation resistance has been deleted.
- Data in Figure 3 are described as showing that surface expression of ICAM-1 was increased after PMA stimulation in all THP-1 derivatives on day 2 and then was nearly reduced to basal level in mock or WT NOD2-expressing THP-1 cells but not in mutant NOD2 THP-1 on day 6. Figure3 shows unevenly matched “y” axis scales in the panels and rows. Also, the marked fluorescent shift in mutant NOD2-expressing derivatives’ isotype controls on Day 6 is concerning and would limit the differences. The bottom row using overlay does not take into account the strong isotype shift especially for R334W.
Answer: The authors used BD CellQuestTM Pro software for analyses of FACS data according to the user’s guide (https://www.natur.cuni.cz/biologie/servisni-laboratore/cytometricky-servis/cellquest-users-guide.pdf). As the y-axis automatically scales to the highest peak in the histogram in this software, its name has been added to the “Flow cytometry” section in MATERIALS AND METHODS. As the authors agree to the Reviewer’s comments regarding the marked fluorescent shift in isotype controls even in Mock cells, the bottom row showing overlay has been deleted from Fig 3 and description of the fluorescent shift in isotype controls has been added in RESULTS.
- The authors mention in Results that they analyzed expression of other surface markers and also did experiments using neutralizing antibodies and provide an assessment of these as a summary of new results, but they do not show data for either. The data must be shown or could be made available as supplementary material but otherwise should not be included.
Answer: The authors agree to the Reviewer’s comments and the statements on the expression of other surface markers including experiments using neutralizing antibodies have been deleted to avoid misleading.
- The legend for Figure 4 should briefly describe the method and delineate the figure. The authors need to consider their flow data in Figure 3, address the isotype shift, and also explain how surface expression of ICAM-1 could be discordant with mRNA expression.
Answer: Brief description of the methods and the results for Fig 3 has been added in the legend and requirement for the quantitative elucidation of the correlation of the surface expression and mRNA expression of ICAM-1 has been discussed in DISCUSSION.
- Abstract: This sentence should be revised to make sense- “Sustained surface ICAM-1 expression was observed in conjunction with this phenomenon, but neither of sustained ICAM-1 mRNA expression or impaired ADAM17 mRNA expression was not observed.”
Answer: The sentence has been revised as “Sustained surface ICAM-1 expression was observed in association with this phenomenon, but neither persistent ICAM-1 mRNA expression nor impaired ADAM17 mRNA expression was observed”.
I believe the manuscript has now sufficiently been revised and is suitable for publication in Children.
Sincerely,
Nobuo Kanazawa
